# GradSimCore: Gradient Similarity based Representative Instances as Coreset

## Abstract

The rise in size and complexity of modern datasets and deep learning models have resulted in the usage of extensive computational resources and a surge in training time and effort. It also has increased the carbon footprint of training and fine-tuning models. One way to reduce the computational requirement is to extract the most representative subset (referred to as *coreset*) that can substitute for the larger dataset. Coresets can thus replace massive datasets to train models and tune hyperparameters, especially in the early stages of training. This will result in a significant reduction of computational resource requirement and reduce carbon footprint. We propose a simple and novel framework based on the similarity of loss gradients for identifying the representative training instances as a coreset. Our method, dubbed as *GradSimCore*, outperforms the state-of-the-art coreset selection algorithms on popular benchmark datasets ranging from MNIST to ImageNet. Because of its simplicity and effectiveness, our method is an essential baseline for evaluating the effectiveness of the coreset selection algorithms. Anonymized codes for implementing the proposed method are provided at `https://anonymous.4open.science/r/GradSimCore-8884`.

## 1 Introduction

Modern deep learning tasks (especially supervised ones) require a lot of data to train the models. They often need extensive hyperparameter tuning to achieve the best accuracy. With fast-growing semiconductor and communication technologies in this digital era, humans have created heaps of digital content (images, videos, text, audio). Massive data indeed serves the data-hungry deep learning to read the complex patterns in the data, which would be otherwise difficult. However, associated challenges include data redundancy, maintenance overhead, and computational requirements to perform learning activities on these digital piles. For instance, the size of various popular object recognition datasets is observed in Table 1. Other application fronts of Artificial Intelligence (AI), such as natural language and speech processing, also have accumulated massive datasets.

Table 1: Popular object recognition datasets and their size

| Dataset | Number of samples |
|---|---|
| CIFAR10 / CIFAR100 (Krizhevsky, 2009) | 60000 |
| MNIST (Deng, 2012) | 70000 |
| Fashion MNIST (Xiao et al., 2017) | 70000 |
| MSCOCO (Lin et al., 2015) | 330000 |
| VisDrone (Zhu et al., 2021) | 300000 |
| IMDB Wiki (Pavlichenko & Ustalov, 2021) | 460723 |
| ImageNet-1K (Russakovsky et al., 2015) | 1431167 |

The large datasets have simultaneously resulted in the complexity of the deep learning models growing exponentially. Millions of model parameters, innovative components, and computationally demanding regularizers make these architectures sophisticated machine learning models. In conjunction with massive training datasets, it has resulted in a significant increase in end-to-end training

time, cost of the required computational resources, energy requirement, and carbon footprint (Killamsetty et al., 2021a).

*Coresets* are weighted subsets of the data which approximate certain desirable characteristics of the full data (e.g., the loss function) (Feldman, 2020). *Coreset* selection is a technique that aims to obtain the most representative training samples from a given large dataset. The primary purpose of this technique is to get a generalization performance on the representative dataset compared to the original training dataset. Training duration and computation requirements for end-to-end training can be reduced drastically if the desired generalization can be obtained.

Naturally, a line of thought based on loss gradients emerged to compose coresets. This set of ideas, referred to as the gradient matching-based methods, work on the expectation that the optimal coreset can approximate the gradients produced by the entire training dataset through a weighted combination. This, in the incremental gradient descent framework, enables the coreset to result in similar updates for the model parameters as the full dataset. This ultimately leads the coreset-based model training to reach a nearby location to the local minima learned by the full dataset.

CRAIG (Mirzasoleiman et al., 2020) suggests finding the optimal coreset by converting the gradient matching problem to maximizing a submodular function and then using a greedy approach to optimize it. The result is a coreset of samples and their importance (weights). GRAD-MATCH (Killamsetty et al., 2021a) constructs an objective for matching the gradients computed over the coreset with that of the complete dataset. They propose minimizing the matching error by casting the objective as a weakly submodular maximization problem and solving it using an orthogonal matching pursuit (OMP) based greedy algorithm. These approaches rely on sophisticated modules such as submodular facility location, OMP, etc. GRAD-MATCH identifies the coreset by selecting different subsets during the course of model training to perform gradient matching. So, to obtain coresets at various fractions of the full dataset, training must be carried out each time. Moreover, in CRAIG, because of the individual weights (or learning rates) on the samples, it may not be straightforward to train the models on the coresets in the mini-batch gradient descent framework. Hence, the training speed gains are not maximal on such coresets.

Our method, *GradSimCore*, is based on the intuition that most samples of a particular class produce gradients in similar directions (particularly in the early stages of the model training). In the case of a class with large intra-class diversity, there can be multiple subgroups of samples based on the similarity of gradient directions. Our method proposes that samples with a strong gradient similarity to many others constitute a representative coreset. In the case of a classification task, which is the focus of this work, to represent all the classes present in the original dataset, we identify representative samples class-wise and combine them into a coreset.

Our approach uses gradient similarity scores to rank the data samples of each class. Based on the desired coreset size, the top-ranked samples from each class are picked to represent the whole dataset. While *CRAIG* uses per-element step size while training from the coreset, our method uses a single step size. We also introduce various steps to reduce the computational time. These steps are discussed in Section 4.

Major contributions of our work can be summarized as:

- Introduces a novel, intuitive, gradient similarity-based method for identifying class-wise representative samples as coreset.
- Thoroughly evaluate the proposed method with respect to multiple object recognition datasets of increasing complexity, different CNN classifier architectures, and cross-architecture generalization study, demonstrating that our method can achieve higher accuracy than state-of-the-art coreset selection methods.
- Overall, this work provides an essential baseline for evaluating the effectiveness of more sophisticated coreset selection algorithms that are forthcoming.

Guo et al. (Guo et al., 2022) have developed an excellent and comprehensive code library named *DeepCore* that implements current popular and state-of-the-art coreset selection methods in a unified framework based on PyTorch (Paszke et al., 2019). They have reported accuracy obtained at various selection fractions on the CIFAR10 and ImageNet-1K datasets. *DeepCore* also provides a framework to interface with other popular object recognition datasets such as MNIST (Deng, 2012), QMNIST (Yadav & Bottou, 2019), FashionMNIST (Xiao et al., 2017), SVHN (Goodfellow et al.,

2016), CIFAR100 (Krizhevsky, 2009), and TinyImageNet. Note that we use the results they reported to identify the state-of-the-art results in a given setting (combination of dataset, coreset size, etc.) and compare our results against it.

## 2 RELATED WORKS

Recently, multiple works have been carried out to perform coreset selection for training deep learning classifiers. Here, we present a few of the most prominent works briefly.

K-Center Greedy approximation (Sener & Savarese, 2018) attempts to solve the minimax facility location problem to select coresets from a large dataset such that the maximum distance between points in the non-coreset and its closest point in the coreset is minimized. Uncertainty-based methods work on the idea that samples having lower confidence may have a higher impact during training than those with higher confidence. Thus, these methods suggest constituting the coresets with the samples with lower confidence. Commonly used metrics to calculate sample uncertainty are least confidence, entropy, and margin (Coleman et al., 2020).

Adversarial Deepfool (Ducoffe & Precioso, 2018) and Contrastive active learning (Margatina et al., 2021) work to find data points distributed near the decision boundary. Gradient matching-based methods work on the expectation that the optimal coreset can approximate the gradients produced by the entire training dataset. Several recent works used gradient-based formulation for the selection of coreset. CRAIG (Mirzasoleiman et al., 2020) selects representative subsets that closely approximate the full gradient. They achieve this by converting the gradient matching problem to optimizing a submodular function using a greedy approach. In particular, they have shown that weighted subsets that minimize the upper bound on the estimation error of the total gradient maximize a submodular facility location function.

Error-based methods try to select training samples that contribute more to the training of the neural networks. Two metrics called the Gradient Normed (GraNd) and the Error L2-Norm (EL2N) scores are introduced by (Paul et al., 2023) that help in pruning significant fractions of training data without sacrificing test accuracy. *GraNd* measures the importance of each sample to the training loss at early epochs. *EL2N* approximates the *GraNd* score, which measures the norm of the error vector, with a higher score indicating higher potential influence. The authors conclude that the images with higher scores tend to be harder to learn (forgettable examples). Their method chooses these forgettable samples as the coreset.

RETRIEVE (Killamsetty et al., 2021c) formulates coreset selection for Semi-Supervised Learning (SSL) as a bil-level optimization problem. This method considers both a labeled set and an unlabeled set to formulate the bi-level optimization problem. It uses a greedy algorithm to select the coreset that minimizes the labeled set loss. GLISTER (Killamsetty et al., 2021b), Generalization-based Data Subset Selection for Efficient and Robust Learning, applies bi-level optimization for supervised and active learning. GLISTER formulates coreset selection as a bi-level optimization problem that maximizes the log-likelihood on a held-out validation dataset.

*GRADMATCH* (Killamsetty et al., 2021a) method follows a similar approach. It introduces a squared $L_2$ regularization term over the weight vectors and uses a greedy Orthogonal Matching Pursuit (OMP) algorithm to select the coreset iteratively. After training on the resulting coreset for a pre-determined number of epochs, the algorithm repeats the coreset construction using the latest iterate.

Work published by (Balles et al., 2021) explores the application of gradient matching for a continual learning setting. Their method tries to select a subset $\mathbb{C}$ from the original dataset $\mathbb{T}$ such that $\nabla L_C(\theta) \approx \nabla L_T(\theta)$ for some loss function $L$ and all parameters $\theta$.

## 3 METHODOLOGY

In this section, we provide a detailed description of our *GradSimCore* approach. Note that we focus on the supervised classification task of object recognition using Convolutional Neural Networks

(CNN) as the chosen classifiers because of their proven effectiveness.

**Notation**:

| | |
|---|---|
| $\mathbb{V} = \{(x_i, y_i)\}$ | The complete training dataset |
| $\mathbb{S} \subset \mathbb{V}$ | Coreset of $\mathbb{V}$ (desired) |
| $\theta$ | Model parameters of the classifier (also represent the model) |
| $\Phi$ | Threshold for neighborhood identification |
| $\mathbb{V}_c$ | Training dataset belonging to class $c$ |
| $g_{x_i}^{\theta}$ | Loss gradients computed at model parameters $\theta$ and data sample $x_i$ |
| $\|\boldsymbol{x}\|$ | $L^2$ norm of $\boldsymbol{x}$ |
| $\mathbb{1}$ | Indicator function |

Our objective is to select a representative subset $\mathbb{S}$ of the complete dataset $\mathbb{V}$ such that model $\theta^{\mathbb{S}}$ trained on $\mathbb{S}$ has a generalization performance close to that of the model $\theta^{\mathbb{V}}$ trained on $\mathbb{V}$. Training deep neural networks is reduced to an empirical risk minimization problem often optimized in the gradient descent framework. In practice, the incremental Gradient (IG) methods, such as Stochastic Gradient Descent (SGD), iteratively estimate the gradient on mini-batches of training data that construct the parameter updates.

Existing gradient-based coreset selection methods such as CRAIG (Mirzasoleiman et al., 2020) and GRAD-MATCH (Killamsetty et al., 2021a) try to find an optimal coreset such that the weighted sum of the gradients of the coreset elements remains within an error margin of the gradients of the full dataset. The objective function can be written as:

$$\underset{w,S}{\arg\min} F\left(\frac{1}{|V|} \sum_{(x,y) \in V} (g_x^{\theta}), \frac{1}{|w|_1} \sum_{(x,y) \in S} (w_x g_x^{\theta})\right) \tag{1}$$

where $\boldsymbol{w}$ is the weights associated with the elements of the coreset, and $F$ is a distance metric.

It can be re-written as:

$$\mathbb{E}_{\theta}\left[\frac{1}{|V|} \sum_{x_i \in \mathbb{V}} g_{x_i}^{\theta}\right] = \mathbb{E}_{\theta}\left[\frac{1}{|w|_1} \sum_{x_i \in \mathbb{S}} w_{x_i} g_{x_i}^{\theta}\right] + \epsilon \tag{2}$$

where $\epsilon$ is the error term having the same dimension as the gradient vector. Note that equation 2 considers the expected value of the approximation error in the parameter space.

Our method identifies the representative samples based on their gradients. Intuitively, data samples whose gradients are similar to that of most other samples best approximate the complete dataset. In other words, these representative samples result in local minima close to the minima achieved by the complete dataset.

Our method thus selects a subset $\mathbb{S}$ of a desired cardinality that approximately covers the whole dataset $\mathbb{V}$. We define the normalized gradient similarity between two samples as

$$\rho(x_i, x_j, \theta) = \frac{< g_{x_i}^{\theta}, g_{x_j}^{\theta} >}{\left\|g_{x_i}^{\theta}\right\| \left\|g_{x_j}^{\theta}\right\|} \tag{3}$$

For each sample $x_i$ in the dataset $\mathbb{V}$, we can measure its ability to represent $\mathbb{V}$ as

$$f(x_i) = \mathbb{E}_{\theta}\left[\sum_{x_j \in \mathbb{V},\, j \neq i} \rho(x_i, x_j, \theta)\right] \tag{4}$$

Using this measure, we compute the suitability of all the samples in $\mathbb{V}$ to become the elements of the coreset $\mathbb{S}$. Essentially, this translates to sorting the dataset samples in the decreasing order of this measure and composing the coreset of a desired size. The exact steps involved in this process are described in the next paragraph.

We start with a randomly initialized model and update its parameters on the complete dataset for a few epochs. We observe in our experiments that generally, 5 to 10 epochs (during which the loss value doesn't plateau) are sufficient. We save the checkpoints of the model parameters after each epoch. For each of these checkpoints, we calculate the gradients of the loss function with respect to the model parameters computed at each data sample. We score the dataset samples based on their ability to represent other samples as denoted in equation 4. We aggregate these scores across the saved checkpoints as an approximation to the expectation over the parameter space.

To improve the time efficiency of our approach, we further modify it into a nearest neighbor search algorithm. The per-sample scores denoted by equation 4 are measured in terms of the number of dataset samples present in close proximity in the gradient space. A nearest neighbor search algorithm finds the number of samples within a given radius from each sample (or, with similarity more than a threshold $\Phi$) and assigns it as its score. We then rank the samples according to their aggregated scores across multiple checkpoints. The top-ranked images are selected as the representative coreset.

The formulation of *GradSimCore* can be represented as below. For a given class $c$, the top ranked samples based on their scores are selected.

$$x_j = \arg\max_{x_i \in \mathbb{V}_c} \sum_{\theta} \sum_{j \neq i} \mathbb{1}(\rho(x_i, x_j, \theta) > \Phi) \tag{5}$$

Algorithm 1 presents our approach more formally.

---

**Algorithm 1** GradSimCore algorithm

---

**Require:** Train set: $\mathbb{V}$; Total epochs: $T$; number of classes: $C$; number of coreset images per class: $N$
**Ensure:** Coreset $\mathbb{S}$

> **for** class $c$ in 1,..., $C$ **do**
>     **for** $x_i \in \mathbb{V}_c$ **do**
>         **for** epochs $t$ in 1,..., $T$ **do**
>             compute $g_{x_i}^{\theta_t}$
>         **end for**
>         $f(x_i) = \sum_{\theta_t} \sum_{x_j \in \mathbb{V}_c, j \neq i} \mathbb{1}(\rho(x_i, x_j, \theta_t) > \Phi)$
>         Store $f(x_i)$
>     **end for**
> **end for**
>
> $\mathbb{S} = \emptyset$
>
> **for** class $c$ in 1,..., $C$ **do**
>     $\mathbb{S} \leftarrow \mathbb{S} \cup \arg\text{sort}_{x_i \in \mathbb{V}_c} f(x_i)[: N]$              ▷ Descending order
> **end for**

---

## 4 IMPLEMENTATION OF GRADSIMCORE

We have implemented the proposed method *GradSimCore* using the PyTorch framework. An anonymized version of the codes for implementing the proposed method is provided at `https://anonymous.4open.science/r/GradSimCore-8884`. Multiple steps are undertaken to speed up the coreset selection process. These are enumerated below.

### 4.1 GRADIENTS W.R.T. THE CLASSIFICATION LAYER

Deep learning models such as ResNet-18 have millions of parameters, and it is impractical to consider the gradient of loss with respect to each of the parameters. It is observed that variation of gradient norm is mostly captured by the gradient with respect to the parameters of the last (classification) layer of the neural network (Katharopoulos & Fleuret, 2019). Similar to the earlier works (Ash et al., 2020; Killamsetty et al., 2021a), we avoid gradient computation of all the model parameters,

restricting to only the final fully connected layer while calculating the score of data samples (equation 4).

## 4.2 Nearest Neighbor algorithm

The complexity of similarity score computation (equation 4) among the samples of a particular class of size N is $\mathcal{O}(N^2)$. Clearly, the computation time increases quickly with the number of samples within a class. Further, the complexity is linear in the number of classes in the classification dataset $C$. Instead of computing $N$ similarity scores for each sample in a given class, our approach finds the number of these $N$ samples that lie within its neighborhood. This important modification saves us nontrivial complexity. We have utilized the radius-based nearest neighbor algorithm implementation from scikit-learn (Pedregosa et al., 2011) with the proposed distance (or similarity) measure mentioned in equation 3. This results in a $\approx 10X$ speed up of the computation time.

## 5 Experiments and Results

We have considered three popular benchmark object recognition datasets used in computer vision with varied complexity (number of classes, image resolution, and intra-class diversity). CIFAR10 (Krizhevsky, 2009) dataset consists of $50,000$ color images of dimension $32 \times 32 \times 3$ belonging to 10 different classes, each class having $5,000$ images. CIFAR100 (Krizhevsky, 2009) dataset consists of $50,000$ training images belonging to 100 classes with 500 training images per class. ImageNet-1K (Russakovsky et al., 2015) is a subset of the larger dataset ImageNet, an image dataset organized according to the WordNet hierarchy. ImageNet-1K consists of 1000 classes, with $1,281,167$ training images and $50,000$ validation images.

We used a randomly initialized ResNet-18 (He et al., 2016) architecture as our base classifier for coreset selection. We report classification (generalization) accuracy for various sizes of coreset expressed as fractions of the complete dataset. However, note that for cross-architecture generalization experiments, we also worked with other popular CNN classifiers, such as VGG16 (Simonyan & Zisserman, 2015) and inception-V3 (Szegedy et al., 2015).

We perform the proposed *GradSimCore* coreset selection as described in algorithm 1. On the resulting coreset, we trained the classifier for 200 epochs starting with a randomized initial seed in PyTorch. For statistical significance, we repeated this training procedure 10 times for each coreset size. We compare the mean accuracy obtained by our method with the best mean accuracy reported by DeepCore.

### 5.1 Results for CIFAR10

A comparison of accuracy values obtained on the CIFAR10 for various fractions of the original dataset are presented in Table 2. The algorithms having the best accuracy at each percentage level as reported by DeepCore are mentioned within brackets against the accuracy value.

Table 2: Comparison of results on CIFAR10 Dataset

| Percentage of Dataset | Max. Accuracy DeepCore | Max. Accuracy GradSimCore |
|:---:|:---:|:---:|
| 0.1 % | 24.3 ± 1.5 (GraphCut) | **28.77 ± 1.26** |
| 0.5 % | 34.9 ± 2.3 (GraphCut) | **39.58 ± 0.72** |
| 1 % | 42.8 ± 1.3 (GraphCut) | **48.00 ± 2.10** |
| 5 % | 65.7 ± 1.2 (GraphCut) | **69.40 ± 0.26** |
| 10 % | 76.6 ± 1.5 (GraphCut) | **78.62 ± 1.05** |
| 20 % | **87.1 ± 0.5 (Random)** | 82.64 ± 0.21 |

To study the robustness of our coreset selection method, we have carried out cross architecture generalization performance comparison at $1\%$ and $10\%$ fractions of the full dataset for three deep learning architectures ResNet18 (He et al., 2016), VGG16 (Simonyan & Zisserman, 2015) and Inception-

V3 (Szegedy et al., 2015) and compared results obtained by our method against results reported by DeepCore. The comparative analysis is tabulated in Table 3 and Table 4.

Table 3: Cross Architecture comparison for the 1% coreset of the CIFAR10 dataset

| Target → | ResNet18 | | VGG16 | | Inception-V3 | |
|---|---|---|---|---|---|---|
| Source ↓ | DeepCore | GradSimCore | DeepCore | GradSimCore | DeepCore | GradSimCore |
| ResNet18 | 42.78 ± 1.30 | **48.00 ± 2.10** | 29.01 ± 3.63 | **47.21 ± 0.95** | 37.54 ± 0.62 | **40.35 ± 0.90** |
| VGG16 | 43.02 ± 1.30 | **46.27 ± 0.33** | 27.47 ± 4.00 | **47.64 ± 0.71** | 37.38 ± 2.09 | **40.48 ± 1.12** |
| Inception-V3 | 42.06 ± 0.69 | **46.89 ± 0.58** | 25.00 ± 3.91 | **47.11 ± 0.66** | 37.26 ± 1.23 | **38.98 ± 0.81** |

Table 4: Cross Architecture comparison for the 10% coreset of the CIFAR10 dataset

| Target → | ResNet18 | | VGG16 | | Inception-V3 | |
|---|---|---|---|---|---|---|
| Source↓ | DeepCore | GradSimCore | DeepCore | GradSimCore | DeepCore | GradSimCore |
| ResNet18 | 76.65 ± 1.48 | **78.62 ± 1.05** | 75.29 ± 1.05 | **78.22 ± 0.29** | 73.94 ± 1.11 | **74.97 ± 0.89** |
| VGG16 | 78.66 ± 0.55 | **78.75 ± 0.16** | 77.91 ± 0.71 | **78.84 ± 0.33** | 76.64 ± 1.25 | 75.92 ± 0.35 |
| Inception-V3 | 75.49 ± 0.91 | **79.68 ± 0.25** | 75.15 ± 1.09 | **78.66 ± 0.24** | 73.69 ± 1.42 | **75.36 ± 0.63** |

## 5.2 RESULTS FOR CIFAR100

We follow the same procedure to select a coreset from the CIFAR100 dataset. We train a randomly initialized ResNet-18 model on the resulting coreset for comparative analysis of accuracy obtained over multiple runs. In each run, the model is trained for 200 epochs starting with a random PyTorch seed. The comparative analysis is presented in Table 5. The algorithms having the best accuracy at each percentage level as reported by DeepCore are mentioned within brackets against the accuracy value.

Table 5: Comparison of results on CIFAR100 Dataset

| Percentage of Dataset | Max. Accuracy DeepCore | Max. Accuracy GradSimCore |
|---|---|---|
| 0.5 % | 8.12 (GraphCut) | **11.95** |
| 1 % | 12.82 (GraphCut) | **18.26** |
| 5 % | 31.6 (GraphCut) | **36.08** |
| 10 % | 41.37 (GraphCut) | **43.34** |
| 20 % | **56.6** (GraphCut) | 50.7 |

## 5.3 RESULTS FOR IMAGENET-1K

We believe the effectiveness of coreset selection algorithms must be tested in the face of complex datasets. Hence, we evaluate our algorithm with ImageNet-1K dataset. Following the similar procedure outlined in previous subsections, we extract coresets of varying sizes using a randomly initialized ResNet-18 model. Similar to other experiments, we train multiple models on the resulting coreset. Each model is trained for 200 epochs with a random PyTorch seed. The generalization performance and comparative analysis are tabulated in Table 6. The algorithms having the best accuracy at each percentage level as reported by DeepCore are mentioned within brackets against the accuracy value.

## 5.4 CORESET IMAGES FOR IMAGENET-1K

In this subsection, we visualize the top and bottom-ranked samples according to our coreset selection algorithm. We chose the 'Zebra' and 'Oscilloscope' classes from the 1000 ImageNet classes. Figures 1 and 2 present the top and bottom-ranked 12 images. The selected samples clearly emphasize the fact that the images with top ranks are representative of the class, while images at bottom

Table 6: Comparison of results on ImageNet-1K Dataset

| Percentage of Dataset | Max. Accuracy of DeepCore | Max. Accuracy of GradSimCore |
|---|---|---|
| 0.1 % | 1.29 ± 0.09 (CAL) | **1.88** ± 0.10 |
| 0.5 % | 7.66 ± 0.43 (GraphCut) | **11.58** ± 0.15 |
| 1 % | 18.10 ± 0.22 (GraNd) | **22.82** ± 0.07 |
| 5 % | **47.64 ± 0.03** (Forgetting) | 43.50 ± 0.41 |
| 10 % | **55.12 ± 0.13** (Forgetting) | 49.51 ± 0.26 |

positions either have some ambiguity in terms of describing that particular class or have images of elements from other classes. As our objective is to select the most representative images from each class, the visualization provides proof for the intuition over which the proposed method is founded. Please refer to the Appendix A for more visualizations.

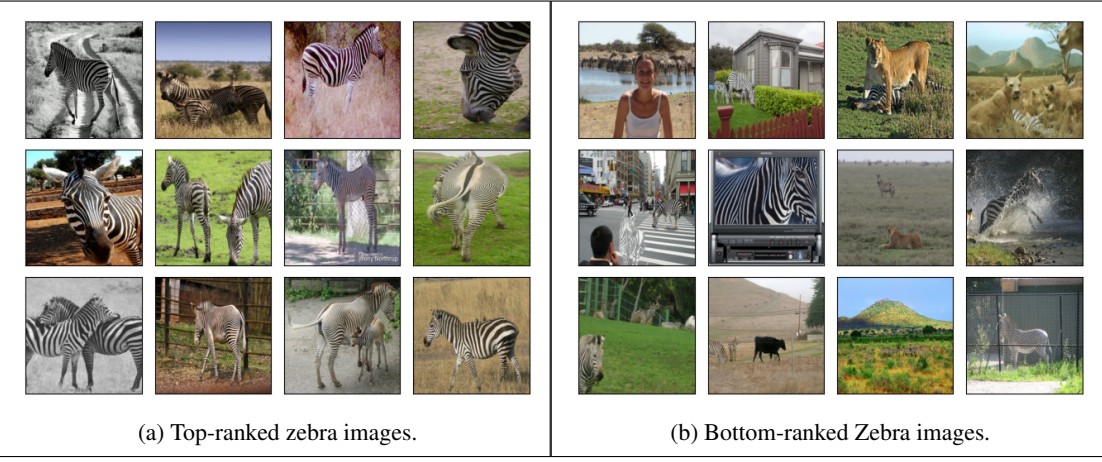

(a) Top-ranked zebra images.    (b) Bottom-ranked Zebra images.

Figure 1: Best and worst ranked images by our method for the 'zebra' category.

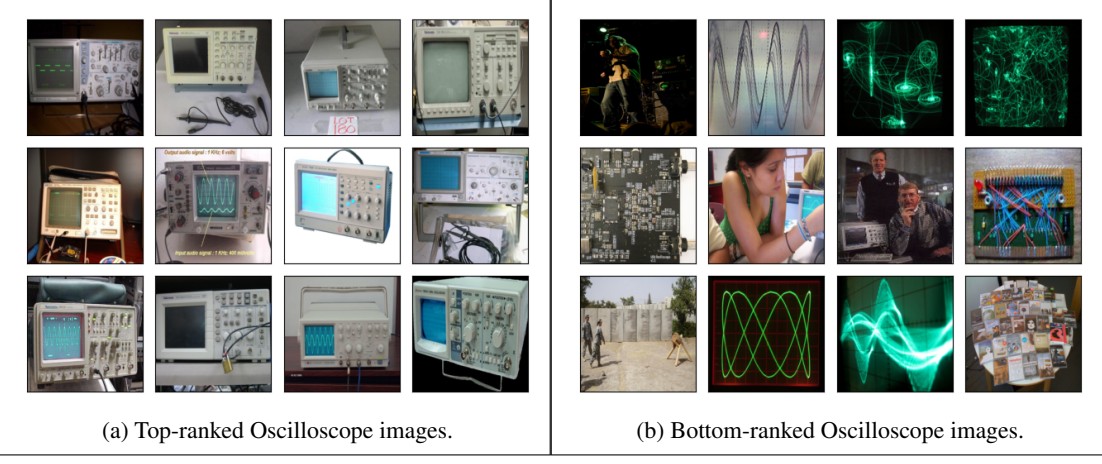

(a) Top-ranked Oscilloscope images.    (b) Bottom-ranked Oscilloscope images.

Figure 2: Best and worst ranked images by our method for the 'Oscilloscope' category.

## 6 CONCLUSIONS

In this paper, we introduced *GradSimCore*, an intuition-driven, gradient similarity-based coreset selection method for identifying representative instances from large datasets. We studied its effectiveness on popular object recognition datasets (CIFAR10, CIFAR100, ImageNet-1K) at various coreset sizes. We demonstrated superior generalization performance of classifiers trained on the resulting coresets. Particularly, we showed that our method consistently achieves state-of-the-art accuracy at the smaller coreset sizes (up to 20% in the case of CIFAR10 and CIFAR100 and up to 1% in the case of ImageNet-1K). We have also demonstrated that our model achieves consistently higher performance in the case of cross-architecture generalization. Thus, we strongly propose our method as an essential baseline to benchmark the forthcoming coreset selection methods. While one can appreciate the simplicity and intuition behind the proposed coreset selection method, it must be studied further. Particularly with respect to the approximation error between the gradients computed by the coreset and full dataset. Inducing diversity in the selected samples along with individual importance weights (or per-sample step size) would further improve the effectiveness of the proposed approach. We consider these aspects for future study.

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

# A    APPENDIX

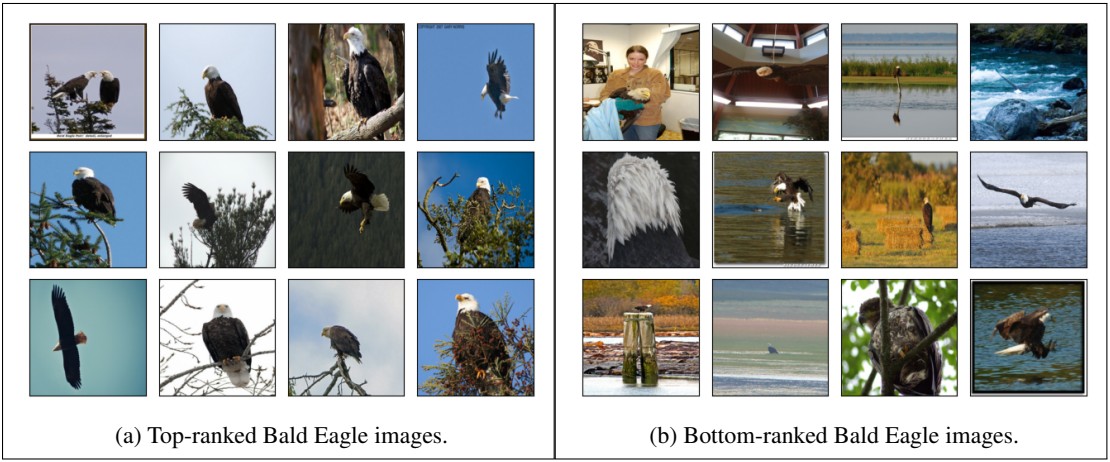

(a) Top-ranked Bald Eagle images.                    (b) Bottom-ranked Bald Eagle images.

Figure 3: Best and worst ranked images by our method for the 'Bald Eagle' category.

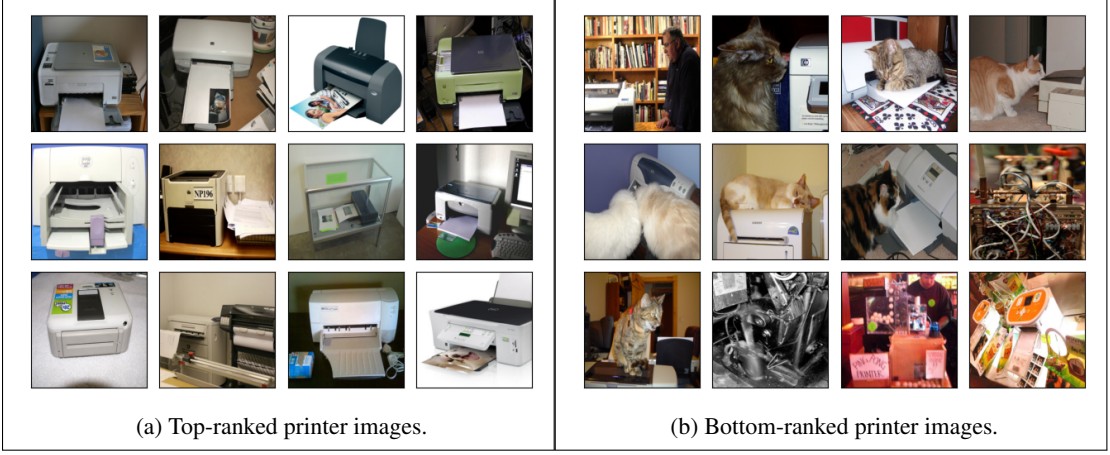

(a) Top-ranked printer images.                    (b) Bottom-ranked printer images.

Figure 4: Best and worst ranked images by our method for the 'Printer' category.

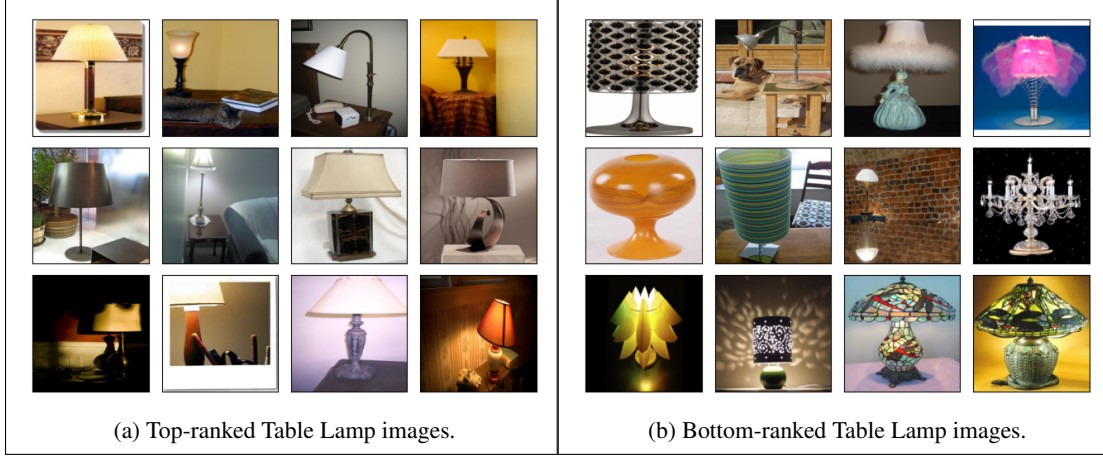

(a) Top-ranked Table Lamp images.                    (b) Bottom-ranked Table Lamp images.

Figure 5: Best and worst ranked images by our method for the 'Table Lamp' category.

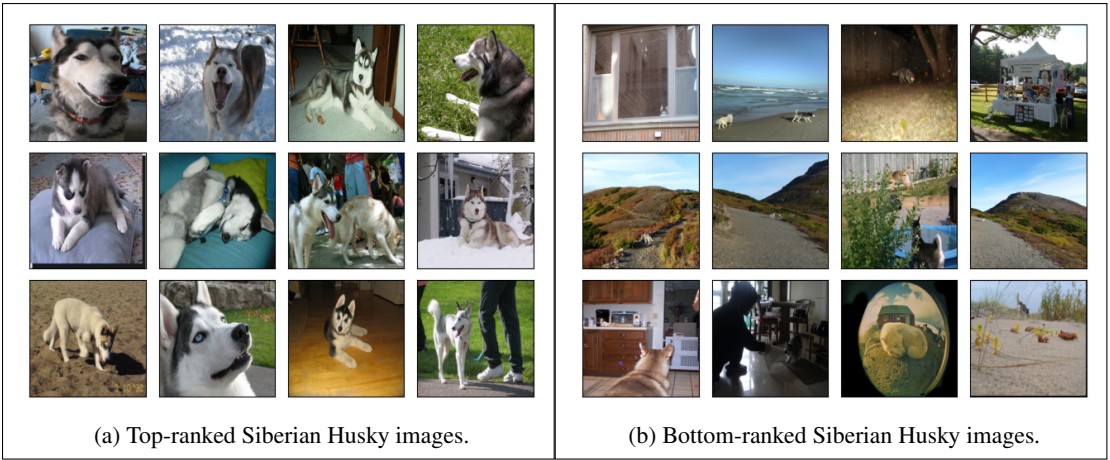

(a) Top-ranked Siberian Husky images.  (b) Bottom-ranked Siberian Husky images.

Figure 6: Best and worst ranked images by our method for the 'Siberian Husky' category.

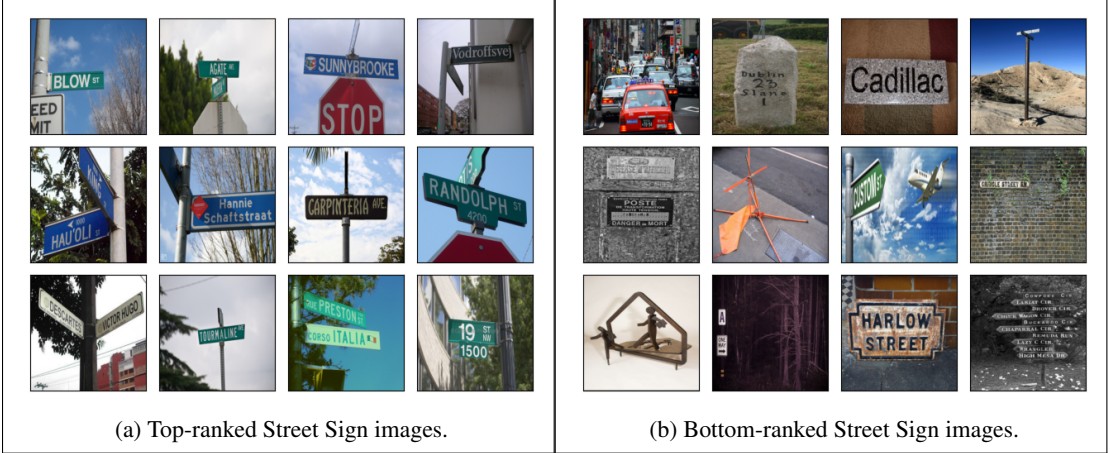

(a) Top-ranked Street Sign images.  (b) Bottom-ranked Street Sign images.

Figure 7: Best and worst ranked images by our method for the 'Street Sign' category.

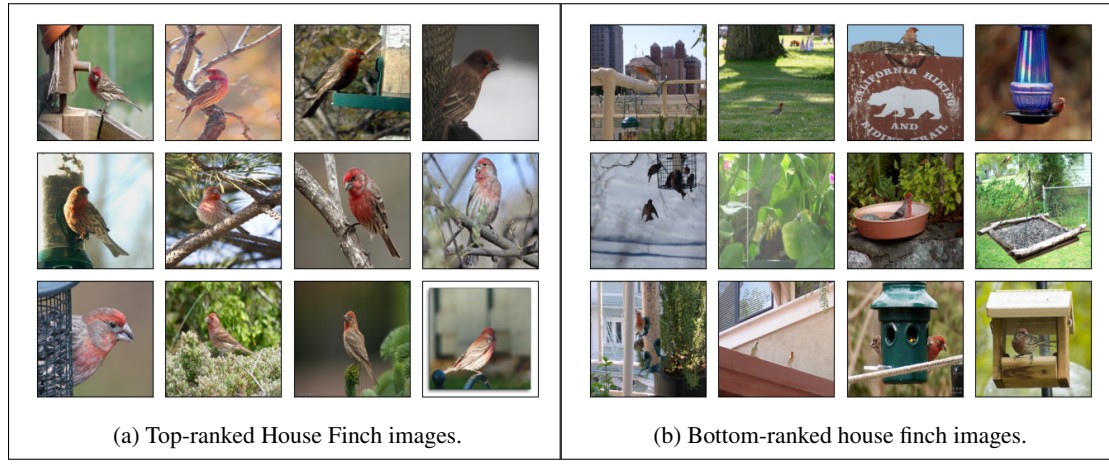

(a) Top-ranked House Finch images.  (b) Bottom-ranked house finch images.

Figure 8: Best and worst ranked images by our method for the 'House Finch' category.

