# OpenReview forum: "GRADSIMCORE: GRADIENT SIMILARITY BASED REPRESENTATIVE INSTANCES AS CORESET"
_ICLR.cc/2024/Conference — Submitted to ICLR 2024_

### Official Review · Reviewer_iu1D · 2023-10-28

**Soundness:** 2 fair
**Presentation:** 1 poor
**Contribution:** 2 fair
**Rating:** 3
**Confidence:** 4

**Summary:**

The paper presents a core-set selection method based on the gradient-direction similarity (cosine distance) at the early phases of training (5-10 epochs). While the underlying idea (and its simplicity) is appealing and first experimental results are promising, the paper does not seem to be in a finalized form. I believe it requires a further iteration to establish the full potential of the method. Length and content-wise, it would fit a densely written 4-page workshop paper very well in preparation for a future main conference paper submission, e.g., for ICML or CVPR. Overall, I think this paper (in the current version) is not refined/detailed enough for the main conference and would require further experimental and comparative work to be competitive with other submitted papers.

Concretely,
1 while several approaches are discussed in related work, the proposed GradSimCore method is only compared to "DeepCore".
2 also this "DeepCore" comparison is not very clear: it is stated that DeepCore is a library for coreset selection. To which algorithm in this library is the method compared to?
3 the manuscript overall seems to be very stretched in the length to barely fill the length requirement of 9 pages. Tables are stretched; Algorithm 1 could fit half the width, The notation table is not necessary. Can be described at shorter length in text, Table 1 could also be described shorter in text.

Content-wise,
1. it is not clear if the method seen as a baseline (called as such in the discussion) or as a new state-of-the-art method (it beats the DeepCore comparison method across several datasets)
2. equation 3 seems to be the cosine similarity. It should be also called as such.
3. equation 4. Why is the expectation used here? Interpreting this (discrete) equation, a simple average 1/N sum would be sufficient and clearer
5. equation 5. How is the threshold chosen? Given the experiments (percentage of the datasets), it seems that the threshold is not used but the samples are rather sorted by f(x) score and the top x% samples are chosen

**Strengths:**

* efficient and effective core-set selection is an important direction that requires further approaches and more research.
* the method is overall intuitive and can be tested/implemented from the paper

**Weaknesses:**

* the paper does not seem to be in its final version (in terms of length, experiments, analyses, number of comparisons)
* the comparisons are very limited, given that many more methods are listed in related work

**Questions:**

1. which DeepCore method was GradSimCore compared to?
2. is this approach applicable to datasets that are gradually collected? I.e., by rejecting newly redundant collected samples that produce a low score? It seems from this version of the paper, that the sample-wise importance scores are calculated on the entire dataset first. Then afterwards, the subset of this dataset is identified. This seems to contradict the motivation of the paper of requiring smaller datasets that are more representative overall (Table 1)

**Details Of Ethics Concerns:**

no ethical concerns

---

> ### Author Response · Authors · 2023-11-22
>
> We thank the reviewer for their valuable time and feedback. We have detailed the responses to their queries below.
>
> - DeepCore library contains the 12 most popular and SOTA methods for coreset selection. They have conducted extensive experiments and reported the best accuracy obtained for various percentage levels of coreset selection for the CIFAR10 and ImageNet-1K datasets. We have utilized their results and proven that our method is able to beat the performance of SOTA methods in the majority of coreset selection percentage levels. We thank the reviewer for pointing this out, and we will update the draft to mention the specific algorithms for each percentage of coreset selection for different datasets.
>
> Q1. Our method is shown to achieve better performance for the majority of coreset percentage levels across datasets and models. However, given the intuitive hypothesis over which the method is constructed and its simplicity, we opine that it acts as a strong baseline for any future coreset methods to compare against.
>
> Q2. We call it “normalized gradient similarity” as it measures the similarity score between normalized gradient vectors. It can also be termed as cosine similarity between normalized gradients.
>
> Q3. We agree with the reviewer that an average would be sufficient. However, the draft presents this notion more formally (i.e., the ranking of individual samples has to happen over the distribution of model parameters).
>
> Q4. Threshold is a hyperparameter that we have chosen based on multiple experiments. The threshold is used to calculate the number of neighbors having a similarity score higher than the threshold value.
>
> **Questions**
>
> Q1. We thank the reviewer for pointing this out; we will update the draft with the specific algorithms that perform best in all the settings.
>
> Q2. No. In the current setting, this method is used for large static datasets and to select a representative smaller dataset. This is a valuable tool in settings such as continual learning where we have a large dataset but are limited by the time available for finetuning the model (to resume catastrophic forgetting).

---

> > ### Comment · Reviewer_iu1D · 2023-11-22
> > **Thank you for the response**
> >
> > Thank you for the detailed responses!
> >
> > DeepCore
> > The main question remains: Which of the 12 DeepCore methods did the authors compare their method to? Always the best method? I.e., a mix of methods depending on which one was most competitive?
> > I would have expected a full table of the individual methods including the random baseline that seems quite strong already in the DeepCore comparisons.
> >
> > Regarding the questions
> > Q1. When I open the attached PDF the tables still state "Max. Accuracy of DeepCore". Is that the revised version? Just stating it in the response here is enough (see raised question above). In particular, since DeepCore comparisons show that random is a strong baseline already. How does this method compare to random?
> > Q2. Thank you for clarifying. In my opinion, a greedy approach where the dataset is gradually enlarged would be more helpful. I generally understand the continual learning argument, but I don't think that time is in practice a limiting factor when fine-tuning models.
> >
> > Overall, my impression that the paper was not finished at submission time remains. I believe it will be a good paper in a future iteration with further extensive comparisons and additional experiments (also qualitative ones)!

---

> > > ### Author Response · Authors · 2023-11-22
> > > **Apologies, the draft is now updated with the SOTA methods in each of the settings**
> > >
> > > We thank the reviewer for sharing their comments. Earlier, we missed updating the pdf of the draft. You may now find the tables updated with specific methods that perform the best in specific settings. To clarify your question, the winner in each setting is a different method; we report that specific top-performing method. We hope this clarifies further. Thank you.

---

### Official Review · Reviewer_1xJD · 2023-10-30

**Soundness:** 1 poor
**Presentation:** 1 poor
**Contribution:** 1 poor
**Rating:** 3
**Confidence:** 4

**Summary:**

The use of gradients as indicators is not originally proposed in this article. The only good thing about this article is that it provides links to the code, beyond that I don't see any good things, whether it's typography, experiments, theoretical proofs, or methodology. I suggest that ICLR can access the GPT-4V interface and directly reject such low-quality submissions.

**Strengths:**

This article is that it provides links to the code

**Weaknesses:**

See the summary.

**Questions:**

No question! I suggest the author should not resubmit a manuscript like this again.

---

> ### Author Response · Authors · 2023-11-22
> **Our work is the first successful attempt to explore the noise-free gradients towards coreset selection**
>
> We thank the reviewer for their valuable time and feedback. We have detailed the responses to their queries below.
>
> - The use of gradients for coreset selection has been proposed by multiple authors (e.g., CRAIG, GRADMATCH). However, the proposed method uses the gradients in a novel and simpler manner to select the coreset. Specifically, the method uses the similarity between the gradients to select the most representative coreset. In other words, the noise-free gradients represent most of the samples of huge datasets. To the best of our knowledge, there is no such attempt made. Please note that no reviewer has pointed out existing works that explore this.
> - We have provided empirical evidence of the robustness and performance of our method against multiple datasets and models. We appeal to the reviewer to notice that often simpler ideas lead to sophisticated algorithms, which, in this case, is sufficiently demonstrated. Going by the reviewer's argument, one has to label all the variants of gradient descent (momentum, Adam, RMS Prop, etc.) as not novel enough (since they all use the basic idea of backpropagation/gradient descent).

---

### Official Review · Reviewer_LXPe · 2023-10-31

**Soundness:** 1 poor
**Presentation:** 1 poor
**Contribution:** 2 fair
**Rating:** 3
**Confidence:** 5

**Summary:**

This paper proposes a novel coreset selection method, GradSimCore, to select the representative coreset from the original large dataset. The novel part of GradSimCore is its new metric to measure the importance of each example. To calculate the importance of an example, GradSimCore calculates and sums the cosine value between the gradient of this example on a certain model and the gradient of other examples on the same model as the importance value. The evaluation results on CIFAR10, CIFAR100, and ImageNet datasets show that GradSimCore outperforms methods implemented in DeepCore for most cases.

**Strengths:**

1. This paper proposes a novel metric to calculate the data importance based on the similarity of loss gradients.

2. The paper compares the proposed method with baselines implemented in DeepCore and shows that GradSimCore outperforms other baselines in most cases.

**Weaknesses:**

My major concerns on this paper are on the evaluation part.

The paper only compares the baselines implemented in DeepCore, which does not include some SOTA coreset selection baselines, like EL2N[1], CCS[2], and Moderate[3]. It is not convincing enough to demonstrate the effectiveness of the proposed method without comparing it with those SOTA baselines.

Besides, the evaluation data in Table 2,5,6 is kind of selective. The settings for all evaluations have “percentage of dataset” of less than 20%. It seems that GradSimCore’s performance drops a lot with a larger percentage of datasets (even with the reported data). It will be helpful to compare the coreset selection performance under various percentages of datasets to better demonstrate the effectiveness of GradSimCore.

[1] Paul, Mansheej, Surya Ganguli, and Gintare Karolina Dziugaite. "Deep learning on a data diet: Finding important examples early in training." Advances in Neural Information Processing Systems 34 (2021): 20596-20607.

[2] Zheng, Haizhong, et al. "Coverage-centric Coreset Selection for High Pruning Rates." The Eleventh International Conference on Learning Representations. 2022.

[3] Xia, Xiaobo, et al. "Moderate coreset: A universal method of data selection for real-world data-efficient deep learning." The Eleventh International Conference on Learning Representations. 2022.

**Questions:**

See weakness.

---

> ### Author Response · Authors · 2023-11-22
>
> We thank the reviewer for their valuable time and feedback. We have detailed the responses to their queries below.
>
> - Coreset selection aims to select a smaller percentage of representative samples to train the deep learning models. Yes, it takes a toll on the accuracy of the corset-trained model. Still, it should be noted that it is an open research area where our method achieves higher accuracy than SOTA models, particularly at smaller sizes.
>
> - DeepCore library contains the 12 most popular and SOTA methods for coreset selection. They have conducted extensive experiments and reported the best accuracy obtained for various percentage levels of coreset selection for the CIFAR10 and ImageNet-1K datasets. We have utilized their results and proven that our method is able to beat the performance of SOTA methods in the majority of coreset selection percentage levels.
>
> - We thank the reviewer for pointing this out, and we will update the draft to mention the specific algorithms for each percentage of coreset selection for different datasets.
>
> **Evaluation**
> - Coreset selection aims to select a smaller percentage of representative subsets. That is why the coreset works report the generalization performance at smaller sizes of coreset. Our method achieved higher accuracy in most of those settings than SOTA models. We notice that our method performs slightly worse at some of the larger sizes than some of the SOTA. Since it attempts to identify the coreset that represents the majority of the dataset, we believe that the proposed method may fall short in capturing the diversity at larger sizes of the coreset.
> - We sincerely thank the reviewer for pointing out the missing baselines and we will update the draft based on these observations.
>
> EL2N and GraNd scores are proposed in the same paper and DeepCore has implemented them. We have achieved higher accuracy than achieved by these methods as reported by DeepCore.
>
> CCS has not reported test set accuracy beyond 90% pruning rate (10% coreset selection rate). They have provided the code base and we will implement to compare performance of our method against theirs.
>
> Moderate have reported test accuracy for 20% and 30% coreset selection only.

---

> > ### Comment · Reviewer_LXPe · 2023-11-22
> > **Thank the authors for the response**
> >
> > Thanks for the additional clarification and discussion in the paper. After I read the authors' rebuttal, I believe that the paper will be better prepared for publication after a more comprehensive comparison with SOTA baselines.

---

### Official Review · Reviewer_z4nU · 2023-11-01

**Soundness:** 1 poor
**Presentation:** 3 good
**Contribution:** 1 poor
**Rating:** 1
**Confidence:** 2

**Summary:**

In this work the authors tackle the problem of coreset extraction from a dataset. This is a key problem in many scenarios from continual learning to meta-learning and HPO. The authors propose a simple technique where they identify samples that produce gradients that are closer to all the other samples (or N nearest samples) from the same class. They construct this representative set to be the coreset for the dataset. By controlling N, they can control the size of the coreset. They run experiments where they benchmark against the best result reported in an open-source library (deepcore) and show that at very low sizes of N, they perform slightly better.

**Strengths:**

The work is very simple and straight forward, particularly so for smaller datasets. Run the dataset through a model for a few epochs to stabilize the training, run every sample and measure the gradients of the last layer, measure a cross-similarity matrix and average it column-wise and sort.

**Weaknesses:**

This work falls short on several areas.
1. There are several assumptions made about the what an averaged gradient would imply or what it means for a sample to produce a gradient that is also closer to other samples, that need a more rigorous mathematical underpinning. It is not often clear or true that samples that produce noise-free gradients are the most representative. In fact, often the opposite is true and has been utilized in a lot of data sampling techniques from hard-negative mining to label softening, just to produce noisy gradients. This is also a reason why regularization techniques such as dropouts work. The authors assumption may only be true in cases where there are some stringent constraints on the loss manifold.
2. The algorithm itself is tremendously computationally expensive. Quick back-of-the-envelope calculations indicate that for complete Imagenet dataset, most modern server-class instances will not be able to even hold the gradients in memory. Computationally, it will be more expensive to identify the coreset, than to run training on the entire dataset.
3. The experimentation is done on small datasets, in smaller models. From the results, it looks like their work only works at low-accuracy levels (or extremely low data sizes). In modern day ML, unless there is parity in accuracy (generalization performance), its not a practically usable model, no matter the constraints. Putting aside the pragmatism, the results show that the authors' approach does not apply at even relatively modest size of coreset.

**Questions:**

The authors compare against deepcore library's best algorithm, but fail to mention which algorithm it is and what were settings used. This does not even convey the basic details. Will it be possible to make a more direct comparison?

---

> ### Author Response · Authors · 2023-11-22
> **Our work is the first successful attempt to explore the noise-free gradients towards coreset selection**
>
> We thank the reviewer for their valuable time and feedback. We provide the responses to their queries below.
>
> **Weaknesses**
>
> Q1. Our work shows empirical evidence for our proposed method that **noise-free garidents can represent a majority of the samples in the training datasets**, thereby formulating a simple procedure for coreset selection. We agree with the reviewer's remarks that there is evidence for the opposite that noisy gradients being explored towards coreset selection (e.g., forgetting, hard sample mining, etc.). Our work, on the other hand, is built on the intuition that the noise-free gradients can also be effective. We have shown that our method achieves higher performance than current SOTA methods across various CNN models and datasets, especially at lower coreset sizes. To the best of our knowledge, there is no such attempt made. Please note that no reviewer has pointed existing works that explore this. We appeal to the reviewer that coreset selection is an open problem, and the proposed methodology provides a different approach to using gradients for coreset selection. Moreover, we reckon that our method is a strong baseline to compare the effectiveness of coreset selection works to compare the effectiveness.
>
> Q2. The proposed method works on the similarity score between the gradients of the loss function w.r.t. the weights of the last fully connected layer of the network. Storing the gradients w.r.t. all the model parameters is not required. Moreover, please note that our algorithm doesn’t consider all the 1.2M images in the ImageNet dataset simultaneously. Rather, it compares class-wise (approximately 1200). Hence, it is not as heavy as perceived by the reviewer.
>
> The proposed method utilizes multiprocessing and nearest-neighbor algorithms to perform this efficiently.
>
> Below, we tabulate the time taken for one epoch processing of ImageNet training: gradient calculation and similarity score computation for that epoch on an RTX-A5000 24 GB GPU. The similarity computation (based on gradient comparison) is performed only once every epoch. Based on these similarity scores, the coreset is selected at the end (of 5 epochs) by ranking the individual images for each.
>
> ----------------------------------------------------------------------------------
> Process                                         |  Time taken
> ----------------------------------------------------------------------------------
> Single epoch full dataset training  |  4.46 hours
> ----------------------------------------------------------------------------------
> Gradient Calculation                     |  3.19 hours
> ----------------------------------------------------------------------------------
> Ranking                                        | 225 seconds
> ----------------------------------------------------------------------------------
>
> Q3. We have provided the results for three different datasets with different class numbers. We have provided results for CIFAR10 ( 10 classes), CIFAR100 (100 classes), and ImageNet-1K (1000 classes). Implementation is done for ResNet-18 (~11 million parameters), VGG-16(~138 million parameters), and Inception-V3 (~24 million parameters) models, which respectfully can not be termed as smaller models).
>
> Coreset selection aims to select a smaller percentage of representative samples to train the deep learning models. Yes, it takes a toll on the accuracy of the corset-trained model. Still, it should be noted that it is an open research area where our method achieves higher accuracy than SOTA models, particularly at smaller sizes.
>
> -  DeepCore library contains the 12 most popular and SOTA methods for coreset selection. They have conducted extensive experiments and reported the best accuracy obtained for various percentage levels of coreset selection for the CIFAR10 and ImageNet-1K datasets. We have utilized their results and proven that our method is able to beat the performance of SOTA methods in the majority of coreset selection percentage levels.
>
> We thank the reviewer for pointing this out, and we will update the draft to mention the specific algorithms for each percentage of coreset selection for different datasets.

---

### Meta-Review · Area_Chair_HcPk · 2023-12-08

**Metareview:**

This paper presents an approach, they call GradSimCore, which selects representative data instances (coresets) of data points using gradient similarity. This is based on approaches like craig and gradmatch.

Several critical issues have been pointed out by the reviewers of this work. Examples include the time taken to compute the coreset being non-trivial, missing baselines, etc. I went through the reviews and responses and I do not think the authors have addressed the concerns adequately. This paper is far from ready for publication.

**Justification For Why Not Higher Score:**

This paper is far from ready for publication.

**Justification For Why Not Lower Score:**

N/A (this is the lowest score)

---

### Decision · Program_Chairs · 2024-01-16

Reject